# Infectious Bronchitis Virus (Gammacoronavirus) in Poultry Farming: Vaccination, Immune Response and Measures for Mitigation

**DOI:** 10.3390/vetsci8110273

**Published:** 2021-11-12

**Authors:** Md. Safiul Alam Bhuiyan, Zarina Amin, Kenneth Francis Rodrigues, Suryani Saallah, Sharifudin Md. Shaarani, Subir Sarker, Shafiquzzaman Siddiquee

**Affiliations:** 1Biotechnology Research Institute, Universiti Malaysia Sabah, Jln UMS, Kota Kinabalu 88400, Malaysia; dr.safiulalambhuiyan@gmail.com (M.S.A.B.); zamin@ums.edu.my (Z.A.); kennethr@ums.edu.my (K.F.R.); suryani@ums.edu.my (S.S.); 2Food Biotechnology Program, Faculty of Science and Technology, Universiti Sains Islam Malaysia, Bandar Baru Nilai, Nilai 71800, Malaysia; sharifudinms@usim.edu.my; 3Department of Physiology, Anatomy and Microbiology, La Trobe University, Melbourne, VIC 3086, Australia; s.sarker@latrobe.edu.au

**Keywords:** infectious bronchitis virus, vaccination, immune system, mitigation strategies

## Abstract

Infectious bronchitis virus (IBV) poses significant financial and biosecurity challenges to the commercial poultry farming industry. IBV is the causative agent of multi-systemic infection in the respiratory, reproductive and renal systems, which is similar to the symptoms of various viral and bacterial diseases reported in chickens. The avian immune system manifests the ability to respond to subsequent exposure with an antigen by stimulating mucosal, humoral and cell-mediated immunity. However, the immune response against IBV presents a dilemma due to the similarities between the different serotypes that infect poultry. Currently, the live attenuated and killed vaccines are applied for the control of IBV infection; however, the continual emergence of IB variants with rapidly evolving genetic variants increases the risk of outbreaks in intensive poultry farms. This review aims to focus on IBV challenge–infection, route and delivery of vaccines and vaccine-induced immune responses to IBV. Various commercial vaccines currently have been developed against IBV protection for accurate evaluation depending on the local situation. This review also highlights and updates the limitations in controlling IBV infection in poultry with issues pertaining to antiviral therapy and good biosecurity practices, which may aid in establishing good biorisk management protocols for its control and which will, in turn, result in a reduction in economic losses attributed to IBV infection.

## 1. Introduction

Infectious Bronchitis Virus (IBV) is an acute and highly contagious respiratory pathogen of chicken that has a major economic impact on poultry stakeholders. The mutable tissue tropism and continuous emergence in various serotypes or genotypes of IBV are prevalent across various geographic regions. As a consequence of the severity and highly contagious nature of IBV infection, it has been implicated in higher economic loss in breeders and layer chickens attributed to reproductive disorders and renal dysfunction. Long-term impacts on the reduction in egg production (up to 20–70%) and loss of egg quality for trade and hatching are observed [1]. IBV infection can cause up to 10–60% mortality over 4–6 weeks of age and poor weight gain in broilers [2,3]. Several studies confirmed that IBV could persist over the long term in tissues and can be transmitted via the faeces of virus-infected chickens at 163–227 days post-infection [1,4,5].

IBV belongs to gammacoronavirus (γCoV) or Group-3 coronavirus (order *Nidovirales*, family *Coronaviridae*) with a positive-sense single-stranded RNA, (+) ssRNA and genome of approximately 27 kb with gene organization: 5′UTR-1a/1ab-S-3a-3b-E-M-5a-5b-N-3′UTR [6]. On the basis of their antigenic cross-reactivity and phylogenetic analysis, CoVs are classified into three major groups [7]: major antigenically related Group-1 of CoVs includes porcine transmissible gastroenteritis virus (TGEV), Feline coronavirus (FCoV) and Canine coronavirus (CCoV). Group-2 comprises Bovine coronavirus (BCoV), Equine coronavirus (ECoV), Murine hepatitis virus (MHV) and Rat coronavirus (RtCoV). Group-3 includes IBV, Turkey Coronavirus (TCoV) and Pheasant Coronavirus [8]. The family *Coronaviridae* is further divided into two subfamilies: *Coronavirinae* and *Torovirinae*. The Coronavirinae subfamily comprises four genera of viruses such as Alphacoronavirus (αCoV), Betacoronavirus (βCoV), γCoV and Deltacoronavirus (δCoV) based on morphology and genome structure [9,10,11]. The first two genera αCoV and βCoV are confined to mammalian CoVs and human CoVs, which are highly pandemic, possess overwhelming spillovers in current history and have raised the significance of public health such as SARS-CoV-2 (etiological agent of COVID-19) and Middle East respiratory syndrome coronavirus (MERS-CoV) [12,13,14]. On the other hand, the other two genera such as γCoV or δCoV usually infects in all kinds of avian species, including chicken, pheasants and galliformes [15]. The virions of CoV are enclosed in a spike-like projection, which is directly involved in the evolutionary processes of IBV by host cell binding, and the represented neutralizing epitope [16]

Vaccinations based on live-attenuated and killed vaccines are produced by referencing local variant serotypes using conventional production strategies that are the only effective controls; thus, it can be applied to manage IBV infections [17]. Even though farmers manage the infection by the implementation of rotational vaccination programs, IBV infection has proven to be recurrent with the emergence of novel variants [18]. Strategies for vaccination have been challenged by the emergence of new IBV serotypes, which include more than 50 reported variants that have been documented globally, even though this results in little or no cross-protection across the spectrum of novel serotypes [19]. The immune response against IBV is directed by humoral and cell-mediated responses against prevailing field conditions as well as the status of vaccination. The innate immune response is pivotal for better management and prevention of IBV infection, but limited information is available [20]. The mechanisms of IBV pathogenesis have been well documented; however, comparative studies of pathological and immunological assessments activated by various IBV serotypes or genotypes are still in progress, and more data will be needed in order to elucidate the precise response to diverse serotypes. This review aims to discuss the advances in current challenges related to vaccine development and viral-induced immune responses. This review also describes various types of immune responses; in particular, the potency of the genetic vaccine may need more exploration in order for future research to underlie the immunological responses to the different types of IBV.

## 2. Epidemiology of IBV

IBV was first reported as a respiratory disease in 1931 [21], which predominantly affected chicks at 2–3 weeks old in North Dakota, USA. However, the causal agent was not detected and was considered as mild respiratory symptoms of infectious laryngotracheitis (ILT). In 1936, Beach and Schalm had proven that novel IBV was completely different from the ILT virus (cross-antigenic determinants) [22]. From time to time, the IBV virus gradually became a major issue in the poultry industry, and researchers were more concentrated on discovering widespread IBV strains in different origins in the world. Genetically related IBV variants have continuously changed in certain regions from the other parts of the world subject to the geographical region [23]. Some variants circulated in most countries, which are currently identified as their individual native variants that will reflect the world’s present condition, such as America, Europe, Asia, Australia and Africa [24]. Most IBV genotypes and serotypes are closely associated with vaccine strains or variants that are very distinct based on geographical areas of each circulating lineage (GI) with respect to the complete nucleotide sequences of S1 (spike) gene [17]. According to phylogenetic analysis, more than six different viral genotypes of IBV are identified with thirty-two distinct lineages (GI-1 to 32), and numerous unassigned recombinants with inter-lineage origin were also identified [25]. The major IBV serotypes were first characterized at Massachusetts in the USA; 4/91 (793B or CR88) from the UK; D274 (D207, D212, or D1466, D3896 and D3128) in Europe; QX-like reported in China; and H120 strains from the Netherlands with several local variants introduced by the local and regional regions by transmission [26]. The strain of IBV-Q1 genotype was first introduced in China from 1996 to 1998; it was genetically and serologically different from IBV classical strains [27]. Most of Mass and 793B are common under GI-(1–13), producing most of the vaccines derived from those strains in many countries. Therefore, vaccine companies are established with QX-based and anti-IBV variant’s vaccines to prevent and control IBV infection [28,29]. Currently, 793B or QX variants of IBV are quickly circulated all over Asia, Europe and Africa and are still undocumented in the USA or Australia since the major Arkansas strains are rarely reported outside the USA. In the last decades, novel IBV variants were isolated from Malaysia and Singapore, few isolates were identical to the Mass serotype and some isolates were more related to China and Taiwan variants [30,31].

## 3. Vaccination

Vaccination is the most effective method for the prevention and control of IBV. Several commercially developed vaccines are available, and their delivery techniques vary depending on vaccine and countrywide local situations. The ideal characteristics of IBV vaccines are as follows: (1) vaccine immunity must be long-term, otherwise re-vaccination is necessary; (2) the selection of the correct antigenic type of vaccination that is specified for wide antigenic variation in order to cover the maximum virulent serotypes; and (3) timing, technique and applications of vaccine according to flock’s status. Expression and delivery systems of various kinds of IBV vaccine with their existing features are shown in Table 1.

### 3.1. Live Attenuated IBV Vaccine

Attenuation of virus or live vaccine is carried out by serial passages of IBV strains in embryonated chicken eggs (ECE) for reducing the required level of virulence. IBV vaccines can either be mild or virulent depending on the level of attenuation. Live vaccines should be covered if those serotypes or strains are circulating in the surrounding region or the predominant antigenic nature refers to strains of serotype-specific vaccines [45,46]. Live vaccines have been proposed to be administered via eye drops, intratracheal or intranasal route, beak dipping and embryonal injection by individual or mass vaccination such as by coarse spraying or drinking water [47]. Current practices are relatively low-cost and result in both local and systemic immunity. Post-vaccination reactions have remained for a few days, including respiratory or clinical symptoms [48,49]. Alternatively, some of the live vaccines have a significant level of residual virulence and tissue damage and high potency to cause airsacculitis found primarily in stress conditions or adverse environmental conditions [50]. Tissue damage caused by attenuated vaccines may proceed to pathological changes or secondary bacterial infections, particularly in day-old chick (DOC) [51]. Embryo vaccination with low virulence is currently practiced in chickens globally in most countries because this technique is a simple method for handling birds administered in DOC at the hatchery. Moreover, high virulence vaccines are used for booster vaccination at days 7 to 10, usually in drinking water for reducing stress. The shortcomings of mild vaccines are reflected in their limited capability to elicit a strong immune response and their specific role in protecting the respiratory system while excluding the ability to protect the kidney and oviduct against nephropathogenic strain or pathogenic strains for the reproductive system. The live vaccines are mostly used to cover mass strains that are strain-specific such as M41, Ma5, H52 and H120 and other monovalent vaccines that are represented as Conn 46, Ark 99, Florida, JMK, 4/72 and D247, respectively. It has been recommended that live vaccines should not be used if vaccine strains collected from other parts of the endemic strains are of different serotypes or have been derived from distant genetic lineages. Studies have revealed that the combination of IBV H52 and H120 vaccine levels can provide long-lasting heterologous cross protection against different IB serotypes [32,49]. There is a caveat with regard to the use of live attenuated IBV vaccines, which can result in the likelihood of recombinant events between the strains employed as vaccines and virulent field strains that may result in the emergence of novel IBV serotypes [18]. In Korea, the protective efficacy of commercial IBV vaccine (H120 and K2 strains) is immunized separately against newly circulating Korean IBV strains, resulting in the K2 vaccine which, in turn, resulted in better protection against protectotypic types of new Korean IBV [52]. The live vaccine is generally applied in the first IBV immunization for early stage immunity without building long-lasting immunity, especially for the region having a higher level of field challenges, with the purpose of local protection in the respiratory tract. Currently, two antigenic types of vaccines are combined to produce wide spectrum serotypes [53]. For this reason, the best vaccination program in broiler chickens is a combination of live vaccination with IB 4/91 or IB88 with Mass type of vaccine in DOC followed by a booster with slightly low attenuated types of Ma5 or H120 immunized at days 10–14 for a wide range of protection against several IBV serotypes (Figure 1). The overall objective of live vaccination is to maintain a basic level of immunity against the challenge posed by IBV in young chicks, especially breeders and layers that are highly susceptible to pathogens in the hatchery.

### 3.2. Inactivated or Killed Vaccines

Inactivated vaccines have provided safety and stimulation for long-lasting, prolonged immunity without vaccination reactions or post-vaccine challenge. The killed IB vaccine is relatively cost-effective compared to attenuated live vaccines and applies either single or two or more combined inactivated antigens or two or more serotypes in bivalent vaccines [54]. Inactivated vaccines are generally produced from IBV-infected allantoic fluid and formulated as inactivated oil-emulsion based vaccines that are applied in layers and broiler breeders before the laying stage. In comparison, the killed vaccines are given by injection or subcutaneous inoculation at 13 to 18 weeks of age of the pullet, which are exposed with live attenuated vaccines. The highest titers have been observed during the middle stage at least 4–6 weeks between the last live and inactivated vaccine during the period of vaccination [55]. Inactivated vaccines do not replicate within host cells compared to live vaccines, resulting in their inability to result in pathological complications. They can be stimulated by a shorter and slow grade of immune response in order to generate the antibody compared to T-cell-mediated immune responses of live attenuated vaccines [56]. Inactivated vaccines usually generate broad defenses against declines in egg production in layers and breeder’s chicken since the potency of live vaccination may not be maintained until the laying stage [57]. The key purpose of applying the killed vaccine is the passive protection to progeny by maternally derived immunity (MDA) from a vaccinated breeder’s hen [58]. The disadvantage of injectable vaccines may result in a reaction at the injection site, causing focal inflammatory myositis ataxia, breast tenderness, fever and ultimately the rejection of the entire carcass for processing due to its physical appearance [59]. The delivery of killed vaccine poses several challenges at large volumes as manual delivery can result in severe stress and increase in mortality as a direct result of accidental deep puncture in the thoracic or abdominal region during the process of vaccination.

### 3.3. Recombinant Vaccines

#### 3.3.1. Viral Vector-Based Vaccines

The foreign genes are expressed by viral vectors, which is an attractive method of gene distribution for vaccine manufacturers. Nowadays, vaccine producers have advanced since they can be exposed to genetic manipulation of the coronavirus genome. Coronaviruses have been exposed to accept and express foreign gene delivery by gene therapy vectors followed by vaccine production [60,61]. For instance, vectors based on an adenovirus gene construct can persist in the host cells without producing any sign of pathology and viral tropism that allows for sustained release of antigens. It is potentially expressed with different types of immunogenic single proteins in vector-based vaccines without a downright virulent organism [62]. Thus, recombinant vector-based vaccines have been developed against IBV, which have promised to enhance immune response and provide long-lasting protection against IBV infection [63]. However, this technique is concerned with reducing the problem related to RNA mutation and reducing the interference of MDA. Therefore, most methods applying constitutive transgene expression showed a specific immune response [64]. Several factors are considered for the selection and design of recombinant IBV vaccines, such as proper protein folding and glycosylation which are vital cellular processes in the host cellular system; in the case where there is a lack of the process causing post-translational modifications, this may change the correct conformation and epitope arrangements that affect the immunogenicity and efficiency of the vaccine [65]. The recombinant adenovirus vector-based vaccine is based on the gene encoding the IBV-S1-glycoprotein, which has been demonstrated to elicit a significant immune response that protects 90–100% chickens against Vic S (serotype B) or N1/62 (serotype C) IBV strains [63]. Several protein antigens have been co-expressed with genes encoding genetic adjuvants, which result in increased immune responses. Alternative studies have revealed a fowlpox virus vaccine containing the IBV-S1-gene, and the chicken interferon-γ gene (rFPV-IFNγS1) increased humoral immunity; therefore, additional cell-mediated immune responses in chickens confer immunity against homologous and heterologous challenges with LX4, LHLJ04XI and LHB IBV strains [66]. In addition, the expression of IBV-S1-gene with chicken IL-18 in a recombinant fowlpox virus vector produced antibody titers, including the higher responses of CD4+ and CD8+ immune system. Likewise, different comparable studies have conducted experiments with the fowlpox virus vector covering only S1 gene and S1 gene plus IL-18. The results showed that the expression of IL-18 with the IBV-S1 gene using a fowlpox virus vector (rFPV-S1/IL18) produced 100% (20/20) protection compared with only 75% (15/20) protection if chickens received a constructed vector containing only S1 without Interleukin [67]. Remarkably, adenovirus vector vaccines have actual beneficial effects in poultry as oral vaccines because maternally derived antibodies could not neutralize the constructed vector, which is confirmed by oral vaccination of mice with adenovirus vector immunized in field experiments [68]. Vector-based oral vaccines may result in a satisfactory transgene-specific antibody response associated with the reduction in stress, handling and side effects [69]. Improvements and modifications are necessary to provide an optimum cell-mediated response, such as dose escalation, nanoparticle coating, combined vectors and translation of the adenoviral genes.

#### 3.3.2. Subunit and Peptide-Based Vaccines

Subunit vaccines comprise a portion of the virus or antigenic principles of the virus, which is cloned and expressed in either bacterial cells, yeast cells or cell lines with the objective of providing a protective immune response without the risk posed by vaccination with the complete pathogen. Subunit vaccines have been prepared from purified viral antigens but are constructed from viral peptides or are a part of the genome encoding for immunogenic epitopes for designing multi-epitope vaccines [70]. Epitope is recognized by neutralizing antibodies that target S1 and N-gene to stimulate or neutralize antibodies and CTL responses [54]. Currently, one group of researchers has developed and tested a novel IBV vaccine based on the multiple epitopes from S1 and N protein coding genes [71,72]. This specific combination of multiple peptides generated a significant number of humoral and cell-mediated immune responses, which resulted in more than 80% protection after challenge with an infectious virus [73]. Another study reported that the lactic acid bacterial system (*Lactococcus lactis*) has been used to deliver peptide vaccines orally; consequently, this technique has been described to be a strong elicitor of mucosal immune response [43]. For example, the birds immunized with twice the dosage of the gene (S1/S2) by S/C resulted in higher protections against IBV challenge than compared to individuals vaccinated with S1 portion [40]. The current approach has applied a complete protein attached as a peptide vaccine to bird’s tissues. Synthetic epitope peptide encompasses the nucleotides encoding S20–S255 as well the polyclonal antibodies against various IBV strains; thus, this signifies its possible applications for a wide spectrum of IB vaccines [74]. These widespread vaccines have been mapped between 19 and 69 and 250 amino acid sequences surrounded by the receptor-binding domain for which its N-terminal plays a vital role in viral recognition and endocytosis [75].

#### 3.3.3. Plasmid DNA Vaccine

Plasmid DNA vaccines consist of a DNA backbone, which carries the antigenic principle that can be transcribed and translated in the host, under the regulation of the appropriate host promoter and protein translational machinery. Plasmid DNA vaccines can be delivered as a monovalent construct, carrying a single antigen regulated by a single promoter, or a multivalent construct, containing one or more antigen coding genes regulated by either a single or multiple promoters [76]. Currently, there is no FDA approved DNA vaccines for applications in commercial poultry farms; however, this expertise has expanded, and essential considerations of many products are still in clinical trials [77]. Yan et al. [39] have revealed the protective efficiency of DNA plasmids encoding IBV-S1 and N or M proteins in the flock by immunizing monovalent or multivalent plasmids. Promising results were obtained after the birds were inoculated with multivalent plasmid DNA vaccine (2×) followed by a single booster with a kill vaccine, with antibody titers in the prime-boost birds being considerably higher than compared to the multivalent DNA vaccine group (*p* < 0.01) with strong concurrent immunity against viral infection. Bande et al. [78] conducted trials with monovalent (either M41 or CR88) and bivalent DNA vaccines encoding the S1 glycoprotein encapsulated within a chitosan-saponin nanoparticle to improve its immunogenicity against monovalent IB-DNA vaccines, which conferred protection against a homologous virus challenge. Another study was conducted on plasmid DNA vaccine with the plasmid construct pDKArkS1 based on the S1-spike genes of Arkansas IBV serotypes and immunization via the in ovo route was applied followed by a live vaccine after two weeks. This strategy elicited strong immunity of up to 100% protection against IBV challenge [79]. Alternatively, the birds inoculated with a single dose of in ovo administered plasmid DNA vaccines without the administration of the live vaccine, which provided a protection of less than 80% after challenge with a virulent IBV strain [80]. Intramuscular vaccination of a liposome-encapsulated multi-epitope DNA vaccine constructed with S1, S2 and N as part of the IBV genome resulted in higher amounts of CD4+, CD3+ and CD8+ cells that resulted in protective immunity in approximately 80% of vaccinated individuals. In order to design for the modification of a vaccine-induced immune response, this is accomplished by a DNA vaccine encoding S gene with a consensus nucleotide sequence gene with the (pVAX1-S_con) [24] IBV N gene or S1-genes with IL-2 [81] or chicken granulocyte-macrophage stimulating factors (GM-CSF) [82]. The studies have shown that S1-encoded DNA vaccines might enhance immune response and showed approximately 95% protection, which is somewhat higher than N-encoded plasmid vaccine [42,83]. The higher immune response was achieved via the efficiency and immune response of DNA vaccines by being used as cationic liposomes carriers [84]. The limitations of plasmid DNA vaccines are associated primarily with the route of administration since the majority of DNA vaccines are delivered via intramuscular injection; therefore, this created limitations for their application in large populations encountered at commercial poultry farms [53]. Currently, most commercial farms use in ovo DNA vaccination at the hatchery to overcome vaccine stress or post-vaccine reaction by IM injection or by delivery via drinking water or mass spraying [80]. A nanoparticle-based DNA drive is good support for protecting the vaccine against enzymatic degradation and improves their efficacy at the mucosal immune level.

#### 3.3.4. Reverse Genetic Vaccines (RGV)

A reverse genetic vaccine is frequently employed to manipulate the full-length genomic cDNAs of viral genomes from RNA virions with the following synthesis of infectious RNA to produce recombinant viruses. This novel technology for operating one or more viral genes is given the potential to develop various modified IBV vaccine candidates [85]. These reverse genetics systems are involved in powerful methods for receiving instance response to the biology of IBV virus, viral transmission and pathogenesis mechanisms [86]. By example, the Beau R-IBV vaccine has developed with three constructed virulent IBV strains by replacing the pathogenic antigenic S1-gene of Beau-IBV strain with an added S1-gene from M41 and European 4/91 pathogenic strains, respectively. These modification constructions generate defensive immune responses against IBV infection without creating a novel Beau-R strain pathogenic [87]. Alternatively, the RG vaccine can be delivered in conjunction with an H120 strain to stimulate a higher level of HI antibody titer than compared to a group vaccinated solely with H120 [88]. The advanced RG IBV vaccines might overcome the neutralization process by the presence of MDA [17]; however, it is challenging to confidently transform the use of reverse genetic-based live attenuated IBV vaccines. Newer generation vaccines can decrease the chances of mutation and viral selection pressure for future analysis and standardization. A summary of different types of IB vaccines with their limitations correlated with the vaccine strategy is shown in Figure 2.

### 3.4. Vaccine Development against IBV

Currently, IBV vaccination programs have gained more attention with respect to the use of low-virulent, live or inactivated killed vaccines with the aim of booster shots at certain times to increase immunity and reduce the antagonistic response of epithelial cells in the respiratory region [89]. However, there is a significant limitation in applying live IBV vaccines because the attenuation of the vaccine is naturally deficient with respect to its capability in stimulating a mucosal immune response [90,91], which is a critical part of controlling IBV infection since killed vaccine can be an option [92,93]. Nonetheless, it was possible that inactivated killed vaccines can be applied to stimulate t mucosal immune responses once combined with several nanoparticles [92]. Different types of IBV vaccines are available in the market, which may vary in vaccine strains, and in nature based on local isolate and recombination in strains isolated from different countries with special legal legislation (Figure 3). Bijlenga et al. [49] described the earlier development of the H strain of IBV containing both the H52 and H120 due to its better capability of heterologous cross-protection against different serotypes of IBV. Further studies have revealed that heterologous IBV vaccines are also more effective for immunizing the 793B-type of variant that has been evidenced to be long lasting with live attenuated IB vaccines and are effectively applied against Italy 02 and QX stain [94,95,96]. The modified live vaccines and inactivated oil emulsions are available for a few serotypes such as Massachusetts, Connecticut and Arkansas in North America. The California strains and Georgia 98 vaccines are collected from the USA. The vaccines (D274 and D1466) are referred to as “Holland variants” and are usually developed in Europe. However, IB H120 based vaccines are used in the entire region of Europe. In the USA, the levels of immunity may depend on geographical sources to provide different levels of immunity and an unusual capability to cross protect against a few heterologous IBVs including JMK and Florida. It was reported earlier that the combined vaccines of IB Ma5 and IB 4/91 variants can provide strong protection against heterologous IBV types. The QX-type live vaccines have been established in Europe with limited use due to its strong potency in the field [4]. The new generation of IBV vaccines has been established against the local dominant D274 strain of live vaccine for future breeding and layer stock. Currently, the Vic-S vaccine is applied in most vaccination programs in Australia for controlling IBV [21]. In Korea, the K2 vaccine may be more effective for the control and prevention of novel types of IBV recombinants as well as variants that are circulated [35]. The chickens vaccinated in ovo are still in progress, depending on the types of IBV strain without killing the embryos [97]. Post-vaccine challenges are found occasionally, causing reversion to virulence in unvaccinated or immunocompromised chickens by a rolling effect that result in high mortality and unpredictable spreading of IBV resulting disease outbreaks [98,99,100].

## 4. Immune Response against IBV

Various defense mechanisms have shown to neutralize a virus for the sake of building up the immune system in the chicken body against IBV. Primarily, the virus enters the body system and is detected and neutralized by the non-specific immune response [101]. Various types of specialized cells are engaged in the immune system that is a critical contributor to the innate and adaptive immune responses shown in Figure 3. Innate immunity is the first line of defense that is involved with physical and chemical barriers and cellular machinery, e.g., phagocytic cells [102], complement [103] and natural killer cells [104]. In contrast, adaptive immunity is characterized with a highly specific response facilitated by T (helper or cytotoxic cells) and B cells (humoral immunity), resulting in a response against infection and the activation of memory cell for recurrent exposure to similar viruses [101,105]. Several innate immune factors, such as heterophils, macrophages, natural killer cells, complement and pattern recognition receptors (PRRs), have been proposed to play a vital role in the induction of immune response against IBV; nevertheless, some factors have yet to be identified [44].

### 4.1. Local Immune System

A vaccine requires a certain period of time in order to elicit a protective immune response in avian hosts. Moreover, passive immunization can induce immunity from maternal derivative’s antibody (MDA), which is particularly supportive during the early stages of life [106]. The structure and function of birds’ immune systems are distinctly different from human immune systems due to their virtue of possessing extra lymphatic organs such as the bursa of Fabricius and the thymus responsible for humoral and cellular immunity, respectively. Furthermore, the birds have carried the secondary peripheral organs of the lymphatic system, for example, the Harderian gland (HG), conjunctiva associated lymphoid tissue (CALT), head associated lymphoid tissue (HALT), gut-associated lymphoid tissue (GALT) and bronchus associated lymphoid tissue (BALT), spleen and cecum tonsils, respectively [107], showed in Figure 4. These assemblies are regularly enmeshed in a chicken’s immune response, especially in the respiratory mucosal system during IBV infection.

#### 4.1.1. Passive Immunity

Passive immunity typically refers to IgG antibodies or maternally derived antibodies (MDAs) that play a significant role in the immediate local protection of chicks for short-term defense against IBV. During the third part of embryonation, IgG is secreted from the yolk and entrance to the bloodstream in order to prevent the virus replication at a point in time. Moreover, the newly hatched chick can receive the IgG level from 5 days of post-hatch, and the continuous circulation of IgG levels in chicken serves as humoral immunity for progeny and improves performance and survival [108,109]. In vaccinated hens, the ovum started to obtain IgG antibody (virus-specific) from blood circulation at 5 days earlier prior to egg laying [110]. The MDAs can persist from a day to several weeks until 3 to 4 months as a maximum depends on the nature and exposure of the virus. Approximately 97% of chicks are protected against IBV infection starting from DOC because of MDAs [98,111]. The level of immunity might gradually decline up to <30% on days 7 of age due to antibody binding and partial neutralization of vaccine viruses as given during short-term protection [50]. The virus can persist until it declines passive immunity to a certain level and starts continued replications until the chicken’s immune system might gradually be boosted to an active immune response simultaneously. Alternatively, neutralizing antibodies can inhibit viral dissemination or replication from the respiratory tract and prevent secondary infection of the reproductive and renal systems. The adaptive transfer of CD8 T-lymphocytes protects chicks against IBV challenge, suggesting a role for cellular immunity as well as in the protection against the virulent IBV strain as a consequence of passive immunity in IBV infection [112].

#### 4.1.2. Active Immunity/Innate Immune Responses

The innate immune system is the first-line defense of IBV infection that can be directly activated during the initial exposure of the virus into the chicken body. The inflammatory response is increased by the flow of blood to assist carrying immune cells to the site of infection [113]. Various kinds of immune cells are involved non-specifically against targeted invasive pathogens. The immune responses are mainly dependent on pathogen-associated molecular patterns (PAMPs) and endogenous damage-associated molecular patterns (DAMPs) through specific pattern-recognition receptors (PRRs) that are exposed to immune cells [114]. The intracellular signaling cascades activated by PRRs result in the transcriptional expression of inflammatory mediators that coordinate with one another to eliminate the virus and infected cells [115].

The immune system is activated by the physical and functional barriers offered by the skin and mucous membranes, which is intended to avoid infection and attack the invading pathogens. Moreover, leukocytes include heterophils, macrophages, NK cells, mast cells, acute phase proteins, basophils and eosinophils that participate in innate immune responses by killing organisms or destroying enzymes and free radicals [17]. The immune cells are associated with macrophages, which have also been activated in the primary stage of infection by the action of the interferon-gamma [116]. Numerous types of innate immune factors are contributed during the preliminary recognition of viral particles through TLR-3 (toll-like receptor) to activate macrophages [117]. In most cases, different types of cytokines and higher levels of interferons (IFNs) have produced airway epithelium infections in respiratory organs causing immunopathology in lung tissue. Cytokines assist in the innate protection of neighboring cells and help mobilize T cell activation and migration of T-lymphocytes to the infected area to influence adaptive responses [118]. The type II interferon is induced by interferon-gamma (IFN-λ) secretion generated by activated NK cells, dendritic cells and CD4+ CD8+ T-lymphocytes. These immune cells represent cytostatic activity, increased antigen presentation on the surface of APCs (macrophages and dendritic cells) and, subsequently, the causes of the expression of MHC-I molecules [119]. Moreover, the activation of CD8+ (cytotoxic) and CD4+ (helper) T cells might be affected in IBV infection directly for virus clearance, resulting in damage to the bird’s adaptive responses [120]. The dendritic cells (DCs) of the chicken act as phagocytic and antigen-presenting cells (APC) by increasing the expression of MHC class I and II molecules on their surface in response to antigens from moving pathogens [121].

Toll-like receptors (TLRs) are a part of pattern recognition receptors (PRRs), which are activated by the innate immune response through sensing-conserved molecular patterns [122]. Various types of TLRs such as TLR4, TLR5, TLR15 and TLR16 have been recognized; therefore, all TLRs are associated with innate sensing with respect to securing the host from viral infections [123]. The significant upregulation of TLR4 has been exposed in IBV infection; however, pattern recognition receptors have a significant role in immune response and defense against other coronavirus infection such as SARS-CoV and mouse hepatitis virus (MHV) [124,125]. Melanoma differentiation-associated protein 5 (MDA5) is the main apparatus in chicken cells that results in the generation of interferon during IBV infection [126].

The Type I interferon is observed in the mucosal lining of trachea, lungs and kidney when an innate immune response is activated during infection; however, the response of stimulation depends on the virulence and host adaptability of the viral strain [118,127]. The primary reactive component of innate immunity against IBV performed the hyperplasia of goblet cells and alveolar mucous glands, resulting in seromucous nasal discharge and catarrhal or caseous exudates in the trachea [128]. A range of immunological approaches are involved, which include macrophage depletion techniques, silica dioxide and related compounds, scavenger receptor manipulation, monoclonal antibodies, clodronate encapsulated liposome-mediated depletion and apoptosis. Macrophages induce various cytokines and chemokines during infectious disease in response to the potential control of infective agents and motivated the adaptive immune responses and cell-mediated immune responses. [129]. Higher levels of proteolytic activity are found in macrophages and transported to insufficient levels of antigen-presenting cells in the host [130]. The depletion of murine macrophages has shown that the number DCs (Dendrites cells) infected in the spleen of chicken increased and gradually increased in the viral genome load in target cells than compared to the macrophages [131]. Silica is mostly used to eliminate monocytes from peripheral blood leukocyte suspension and used to remove macrophages from immune cell populations in vitro [132]. The monoclonal antibody is an alternative method for targeting macrophages for depletion against macrophage-specific surface markers, as used in the target group of diversity CD11b/CD18 integrin, and induces macrophage depletion by complement-mediated lysis [133]. It has been confirmed that IBV induces apoptosis in the advanced stage of infected cells; the Bcl-2 family is controlled by IBV-induced apoptosis [134]. Fung et al. [135] reported the role of unfolded-protein response (UPR) in IBV Beaudette-induced apoptosis and pragmatic induces ER stress in infected cells that triggered the IRE1a-XBP1 passage in the later stage of infection.

### 4.2. Adaptive Immune System

#### 4.2.1. Humoral Immunity

The humoral immune response is triggered by the activation of IBV-specific antibodies to inhibit viral replication and virus circulation in the blood steam or hinders viremia from the trachea, kidneys and oviduct [44,136]. The main function of humoral immunity is to generate antibodies at a certain level through plasma cells with or without the response of T helper cells. Long-term protection against IBV infection might need stimulation of an effector that is memory cell-mediated, which has been described in several studies on the systemic and local humoral immune response to IBV vaccination [3]. However, naive B cells carry surface immunoglobulin and MHC class II molecules on the surface of their antigen-presenting cells (APCs) to form a membrane-spanning bond in order to stimulate the antigen-specific immune response [137]. The immunoglobulins have been identified at days 4–5 after post infection and reached their potential peak at days 21 [138]. Generally, the detection of humoral response to IBV infection is measured by ELISA, HI or VN serological tests by using serum [139].

The antibodies’ response was confirmed in serum samples or tracheal swabs, and lacrimal secretion after successful IBV vaccination [140]. The study found that systemic (IgM and IgG) and mucosal (IgA) immunoglobulins were responsible for inhibiting the virus in infected chickens [141]. Moreover, IgA is the most common immunoglobulin that plays a vital immune function in mucous membranes and tracheal or other mucosal points of viral invasion. On the other hand, IgM is detected at the early stage of infection at 1–5 days of infection, subsequently reaching the peak level at 8–10 days post infection, followed by gradually disappearing at 18 days after infection [142]. The anti-IBV IgG can be identified very early as soon as at 4 days after infection and attains the maximum peak after day 21, following which the higher titers are maintained in the sera for several weeks [143]. Even though humoral immunity directly contributes to the clearance of IBV, except in the cases of IBD (Infectious bursal disease) and CAV (Chicken infectious anemia), infections can present themselves as respiratory symptoms that interrupt the targeted B and T lymphocytes, causing delays in the development of IBV-specific IgA in tears and the interruption of IBV clearance [56,144,145].

#### 4.2.2. Cell-Mediated Immunity (CMI)

In chickens, CMI is one of the essential immunoregulatory weapons during IBV infection, especially for aiding the clearance of viruses, decline of infection and reducing virus shedding and vaccine development [146]. The evaluation of cellular arms is performed by lymphocytic transformation assays, cytotoxic lymphocyte activity [147], delayed-type hypersensitivity [148] and natural killer cell activity [149]. Histological lesions of CMI responses are performed by T-cell infiltration in the respiratory and renal tissues of IBV-infected chickens [150]. The experimental studies have shown a positive relationship between lymphoproliferative responses and resistance to challenge infection [39]. Alternative studies have been published on mouse monoclonal antibodies (mAbs) that differentiate between T-lymphocytes and are used to evaluate the role of T-cells in viral clearance [151]. N and S genes have a specific protein response associated with the stimulated virus-specific protective immunity of CTLs, which is characterized by the reduction in viral load and clearance of the virus from circulation [152,153]. A marked increase in CD4+ and CD8+ T-cells has been described as the recombinant S1-gene associated with the induction of cellular immunity of specific IBV vaccines [154]. Guo et al. [155] reported that IB vaccination with N gene-based DNA vaccine significantly increased the number of CD4+ and CD8 + T cells in peripheral blood mononuclear cells (PBMCs). The existence of CD8+ cytotoxic T lymphocytes (CTL) signifies an essential relationship for reducing infection and resembles a decrease in clinical signs by the action of major histocompatibility complex (MHC), and lysis is facilitated by CD8+CD4 cells [156]. Consequently, the major histocompatibility complex organized cytotoxic T lymphocytes (CTL), and the cytokine activities of chickens participated during the early stages of IBV infections [157]. Several studies have been conducted on tracheal immunity induced by live vaccines, and they found that all vaccines induced significantly higher expression of CD4+ and CD8+ compared with unvaccinated birds using a nephron-pathogenic IBV strain [44]. In the following year, other studies have reported that CD4+ cells are recruited into the trachea earlier than CD8+ on 5 dpi (days of post-infection) [158], which agrees with the findings by Kotani et al. [159] who recognized that the frequency of CD4+ and CD8+ cell numbers significantly peaked at 5 dpi when using a virulent IBV strain. In contrast, studies reported that CD8+ cells were recruited into the trachea earlier than CD4+ cells after infection with virulent 793B [160] or live attenuated IBV vaccine [44] or a combination of live attenuated vaccine with a booster dose of an inactivated vaccine [161]. CD8+ memory T cells have greatly protected the newborn chicks from acute IBV infection at 4 dpi and mild clinical symptoms show at 5 dpi [156]. Even though the adoptive transfer of CD4 + T cells could not be significantly protected in the early stage of IBV infection, primed αβ T cells carrying CD8+ T cells are critical in protecting chicks from IBV infection [152]. Chhabra et al. [44] reported that the protection against Q1-IBV strain changes the quantity of CD4+ and CD8+ cells in the trachea using immunohistochemistry. The results showed that the overall patterns of CD8+ cells are dominant compared to those of CD4+ cells in the two vaccinated groups. The kinetics of CD4+, CD8+ and the IgA-carrying B lymphocytes in the trachea are shown in Figure 5B–D) compared with control Figure 5A in vaccinated groups as differences may have a close relationship with the IBV-specific strains.

#### 4.2.3. Mucosal Immunity

The mucosal immune responses are a more advanced technique for respiratory viruses and show ongoing progress in the development of mucosal vaccines [162]. Mucosal vaccination is regularly applied in the poultry farm because of its cost-effectiveness, effectiveness and reliability of the assay to immunize the large numbers of birds that stimulate local and systemic immune responses [163,164]. Moreover, the IB virus can also replicate in Harderian glands with conjunctiva stimuli designated as mucosal T-independent IgA response to IBV vaccination [44]. van Ginkel [165] reported the IBV-specific IgA and IgM cells secreted from the Harderian Gland by the response of IBV infection identified using enzyme-linked immune-spot forming test (ELISPOT). In addition, several studies have described IBV-specific IgG and IgA that are identified in tears, tracheal and oviduct and duodenal and cecal contents using class-specific monoclonal antibodies in enzyme-linked immunosorbent assay [154]. Generally, live vaccines might be given in the mucosal epithelial via oculo nasal or by spay, which induces local protection faster and significant cellular immune response after reception by the head-associated lymphoid tissues (HALT) followed by antigen-presenting cells. This mucosal immune response is connected with lymphoid development with immune responses of HALTs and the subsequent induction of CTL response [51]. The immunization of the H120 live attenuated vaccination with a higher level of IgA detected in the mucosa acts as local protection against IBV infection [154]. Moreover, Wang et al. [166] reported the gene transcription profile applied in tracheal epithelial cells after three-day infection of chickens with an attenuated IBV-Mass strain; the results established that the varieties of innate immunity and helper T-cell-type-1 based on adaptive immunity are triggered in the host defense mechanism by faster clearance of viruses from the local infection. Recently, Lopez et al. [92] developed nanoparticle IBV-CS vaccines encapsulated in chitosan applied in the oculo-nasal route, inducing significantly increased mucosal immune responses and faster induction of anti-IBV IgA isotype antibodies with IFNγ gene expression at 1 dpi in mucosal sites.

## 5. Potential Mitigation Approaches in Controlling IBV

### 5.1. Limitation for Controlling of IBV in the Poultry Farm

The limitations of controlling IBV are multifactorial, including age of birds, breed, nutrition and management status; and variant or potential use of a live vaccine of IBV against virulence strain. Moreover, mutation (10^−3^ to 10^−5^ errors per replicating cycle) and recombination are the modern instruments of coronavirus that may be responsible for the outbreaks of different strains of IBV [167]. For this reason, IBV may result in developing genetically variable species by increasing their capability of tissue tropism and faster response to alterations in selective pressures [168,169]. As a result, continuous emergence of new IBV serotypes and/or strains often results in resistance to vaccine-mediated protection. The failure of vaccines is one of the vital reasons for the IBV challenge since it may result in an inability to defend chickens against more than one strain or serotype of a virus [170]. The best results were obtained from field studies using live attenuated low virulence IBV vaccines in broiler chickens via spray route followed by booster vaccines (virulent) in drinking water at 7–14 days onward for long protection. However, layer chickens might be vaccinated with live attenuated at an early stage and might receive further immunization with a killed vaccine near the start of laying [55,171].

### 5.2. Issues Related to Antiviral Therapy

The methods of inhibiting coronavirus replication included targeting the main proteinase of coronaviruses that is responsible for virion replication by using antiviral drug therapy. The use of cysteine–proteinase inhibitors in cell cultures has inhibited the replication of many coronavirus species, including IBV [172]. Another potential antiviral drug target is used in viral RNA synthesis, such as the surface glycoproteins used in the fusion of the virus with the host cell membrane. The use of the fusogenic inhibitor enfuvirtide inhibited human immunodeficiency virus (HIV) replication by interfering with the conformational rearrangement of structures within the carboxy-proximal region of the S protein [173]. However, researchers are still far from developing an antiviral drug that is safe, effective and applicable across a wide spectrum of coronavirus species. There is an urgent need for antiviral therapy to develop effective disease control strategies to mitigate the effects of IBV infection in the poultry industry.

### 5.3. Biosecurity and Control of Disease

The basic requirement of biosecurity is to improve primary and secondary containment by implementing control measures such as limited or controlled site access, providing boots and uniform, farm equipment and litter sanitation, foot-baths and wheel baths, which may all need to be set up at all entrances inside/outside where the maximum truck enters the farm. Setting up all biosecurity plans routinely during an emergency outbreak in order to reduce the greater risk of IBV spreading is enforced. Cleaning is the most hygienic measurement and is a significant part of biosecurity for reducing the level of virus infection and transmission. Cleaning should be categorized as dry or wet, which can be applied with an accurate concentration of proper sanitization and disinfected for reducing residual virus contamination. It is highly recommended that similar disinfectants should not be used more than the last 6–12 months, resulting in the development of resistance. Moreover, the downtime between consecutive bird houses is recommended to be at least 10–14 days, except for the multi-age flocks that are challenging and may need to follow strict biosecurity measures for movement and equipment among the flock. Several factors aggravate IBV infection, such as the bird’s strain, age of birds, nutritional aspect and other environmental stress, e.g., ammonia levels, ventilation, temperature and humidity [47,155]. Additionally, the distance between the two nearest poultry farms should not be permitted as it is difficult to free from IBV infection due continually rolling the virus inside the farm. Moreover, with effective containment, the transmission of the virus can be mitigated by an e-vaccination, which is one of the most common practices adopted at commercial poultry farms. Vaccination programs should develop based on local isolation of IBV strains that is recommended for better protection against field strains [174]. For control and prevention of IBV, other respiratory diseases may need proper monitoring and control in order to decrease potential damages to the respiratory tract, which is a predisposing factor to the entrance of IBV. The severity of the IBV disease in the early age of birds can be managed by reducing extreme environmental ammonia and maintaining accurate brooding temperatures. Furthermore, the “all-in/all-out” management policy should apply to prevent the spread of IBV infection between the inter-groups of chicken farms.

## 6. Conclusions

IBV has circulated in the world almost 90 years apart as it was first isolated in early 1931. Currently, the live and killed vaccines followed by booster immunization have been the ideal approaches in IBV vaccination programs; however, the outbreaks of IBV in poultry farms have become persistent, which poses a significant problem to commercial farming. The emergence of new serotypes and variants has necessitated the development of new strategies for the management of IBV and the engineering of new vaccines to address novel serotypes. The innate immunity against IBV virus is generally weak and needs an advanced understanding of the immune responses to IBV. The vaccine technologies have been frequently expanded based on genetic experience and attempts to apply a broad-spectrum or new generation of genetic vaccines based on virulent viruses initiating the field challenge, which is the best method for the control of IBV. This review outlines the present situation of IBV control, including vaccination, immune status of birds and current developments in vaccine technology, for inducing cross protection against multiple IBV serotypes in different regions of IBV that help effective management in the future.

## Figures and Tables

**Figure 1 vetsci-08-00273-f001:**
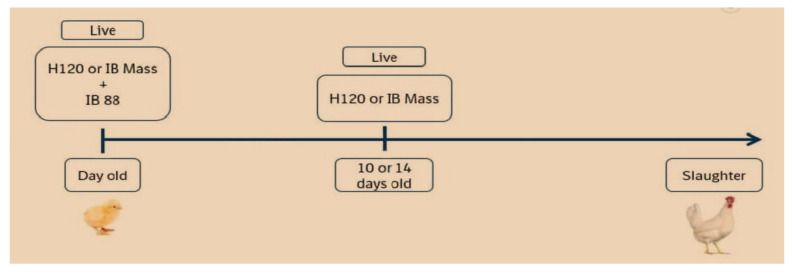
IBV live attenuated vaccination program in broiler chicken with different serotypes widely used in challenge areas for providing stronger protection against several IBV serotypes.

**Figure 2 vetsci-08-00273-f002:**
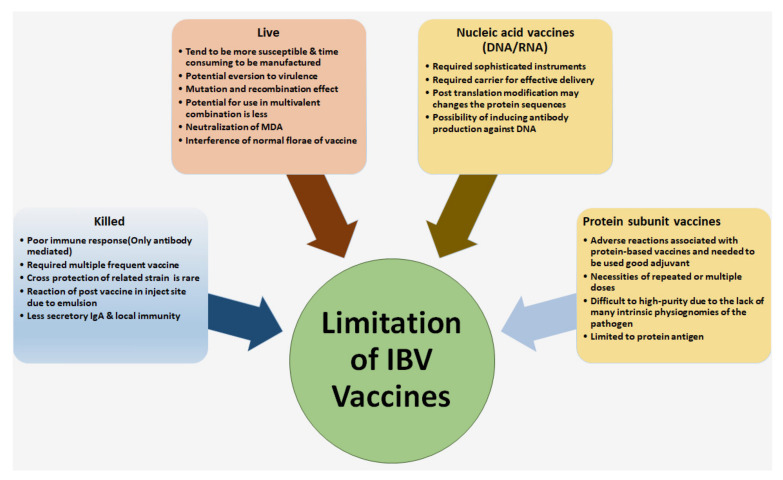
Different types of IB vaccines with limitations correlated with the vaccine strategy and effectiveness.

**Figure 3 vetsci-08-00273-f003:**
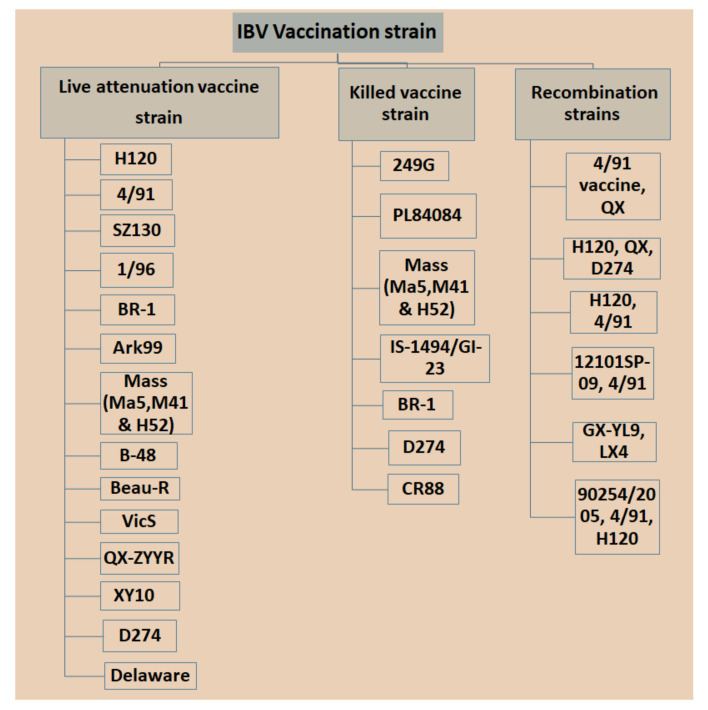
Different types of IBV vaccines are manufactured in the world based on the specific strain.

**Figure 4 vetsci-08-00273-f004:**
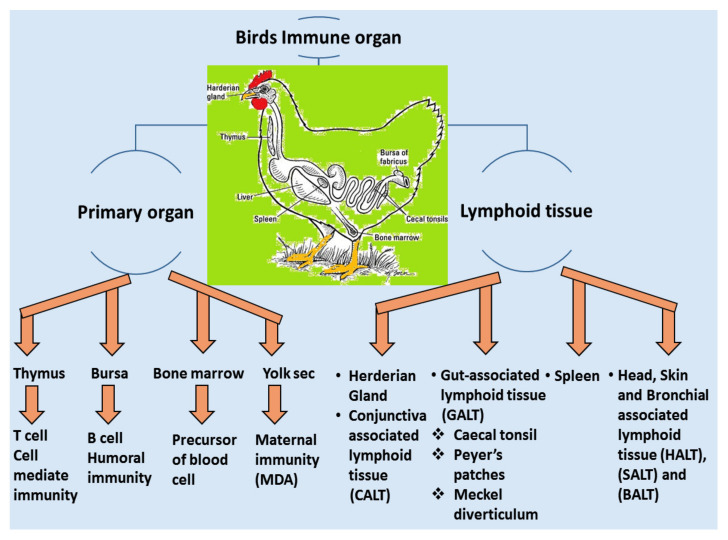
The primary and secondary lymphoid organs enmeshed in the immune complex where the mature B and T cells are transferred from primary or lymphoid tissue for the development stage term as immune movement or immune peripheralization.

**Figure 5 vetsci-08-00273-f005:**
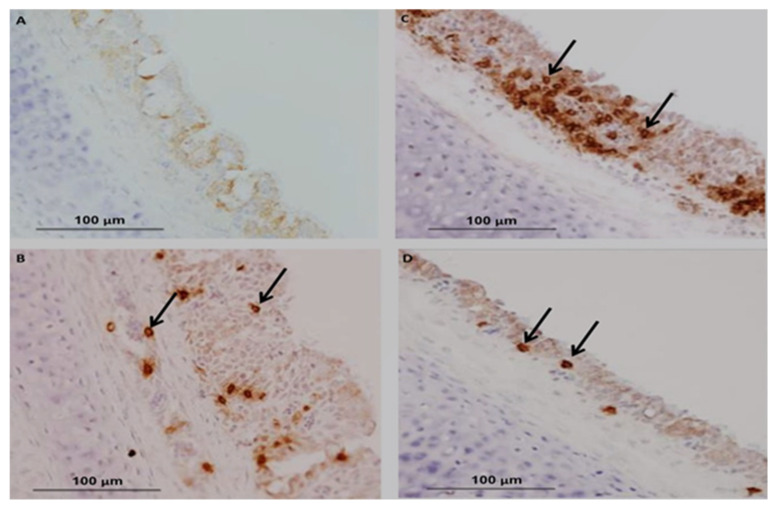
Immunohistochemically found in Tracheal mucosa as control (**A**), CD4+ cells (**B**), CD8+ cells (**C**) and IgA-bearing B cells in (**D**) at day 28 of age. Chickens vaccinated with live H120 or combination with CR88 at day 1 and subsequently with booster vaccinations. Arrows specify the positive immune cells (Magnification ×400) (Reprinted with permission from ref. [44]. Copyright 2021 American Society for Microbiology).

**Table 1 vetsci-08-00273-t001:** Routes of delivery of various IBV vaccines with their associated characteristics.

Name of the Vaccine	Route of Delivery	Characteristics
1. Live attenuated IBV or	Aero nasal spray	Serial attenuation of virulent IB strain for weakened
Live IBV vaccines	In Ovo route	virulence [32,33].
	Orally	
	Subcutaneous (S/C)	
2. Killed or inactivated	IM injection	Inactivated by chemical treatment or heat treat to kill the
IB vaccines	S/C	virulence of strain [34].
3. Viral Vector vaccine	In ovo route	Recombinant rNDV/APMV-2 expressing the S protein of
		IBV strain Mass-41 (rNDV/APMV-2/IBV-S) [35].
4. DNA vaccine	Mucosal/Orally	IBV-DNA vaccine carrying S1-protein and/or N-protein constructs
	IM injection	the respective vector [36,37,38,39].
	Intranasal	
	In ovo route	
5. Recombinant protein (sub-unit)	Intraocular-nasally IM injection	Water-in-oil emulsified recombinant S-ectodomain protein [40].
		Second heptad repeat (HR2) region of S protein were
		repeatedly co-displayed in the Self-assembling
		Protein Nanoparticle (SAPN) [41].
6. Multi-epitope-based	Oral	Using attenuated S enterica serovar Typhimurium strain [42].
peptide vaccine	Mucosal	Recombinant DNA: The EpiC gene was presented in
(Lactococcus lactis bacterial system)	Intranasal	Lactococcus lactis NZ3900 with 3 recombinant strains expressing EpiC gene [43].
7. VLP-based IBV vaccine or	IM immunized	Efficient mucosal immune response [44]
chimeric VLP vaccine		

## Data Availability

The study did not report any data.

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
