# Peer review of "Infectious Bronchitis Virus (Gammacoronavirus) in Poultry Farming: Vaccination, Immune Response and Measures for Mitigation"

_vetsci, 2021, doi:10.3390/vetsci8110273_

Round 1

Reviewer 1 Report

In the manuscript of Bhuiyan et al. the authors provided comprehensive review of literature on vaccination and host immune responses against infectious bronchitis virus. This is economically important pathogens of poultry. Since the immunology section of this manuscript is big and vaccination is a part of mitigation measures, I suggest changing the title of the manuscript to, “Infectious Bronchitis Virus (Gamma-coronavirus) in Poultry Farming: Immune Responses and Mitigation Measures.” Also, after the Introduction, I suggest starting with the Immune responses section followed by Potential mitigation approaches in controlling IBV. Vaccine Development (currently section 4.2) needs to be combined with the Vaccination section. English needs editing. Some lines are difficult to understand.

Further Comments:

Line 11 – there is no affiliation #3 in the authors list.

Citation  - this paragraph refers to another article of this group.

Line 51 – change “SARS-CoV-2 (COVID19)” to “SARS-CoV-2 (etiological agent of COVID19)”

Line 53 - needs to be “usually infects all kinds of avian species, especially chickens …”

Lines 55-56 – too many “using” in this sentence.

Lines 72-75 – bad English

Line 78 – What is “immune-pathogenesis response” ?

Line 84 – words “good feature of IB vaccinations” needs to be changed.

Table 1 – needs to be formatted to be on one page. “Delivery system” must be Routes of vaccination. Change “Ovo” to “In ovo”. Viral vectored , recombinant protein (sub-unit) and DNA vaccines must be separated    

Line 111 – “The difficulties of mild vaccines…” must be “drawbacks” or “disadvantages”

Line 117 – “It has been highly suggested…” may be “strongly suggested”

Line 120 –“ the combination of IBV H52 and H120 vaccine levels” not clear.

Line 124-125 – bad English

Line 152-154 – bad sentence

Line 158 – Spell out MDA

 Line 158 -161 – bad sentence

Line 177-178 – bad sentence

Line 185 – “is enclosed in” must be “includes”

Line 191 – “cell-mediated immune responses are protected in chickens”. Bad English

Line 206 – “and transaction of adenovirus gene.” What is transaction?

Line 236 – “in the flock by immunizing monovalent or multivalent in plasmids” Do not understand.

Line 237 – “The finding of results shown…” Bad English.

Line 240-244 – Bad sentence.

Line 245-247 –Bad sentence.

Line 249-252 –Bad sentence.

Line 258-259 – Bad sentence.

Line 265-267 – Bad sentence.

Figure 2 – I am doubt that production of live vaccine is more time consuming and costly than killed vaccine. What is “interference from normal florea of vaccine”? What is “Genetic vaccine”? Is it DNA vaccine, viral vectored vaccine, sub-unit or vaccine generated by reverse genetic technique? Meaning of “limited to protein antigen” is not clear.

Line 377 – Reference 95 is for human TLRs. What about references for chicken?

Line 410 – Mention the main targets for VN antibodies.

Line 420 – “days 4-5 after post infection”

Line 433-437 – Bad sentence.

Figure 4 – Do you have permission to present this figure from reference 27?

Line 516 – “the modern instruments..” ?

Line 526 – Bad English

Line 528 – 563 – this section needs to be combined with vaccines.

Figure 5 – What is the difference between Live vaccine and Live Modified strains?

Line 600 – Bad English

Line 610 – Bad English

Line 614 -  “to be stably replicated” ?

Author Response

Reviewer #1

In the manuscript of Bhuiyan et al. the authors provided comprehensive review of literature on vaccination and host immune responses against infectious bronchitis virus. This is economically important pathogens of poultry. Since the immunology section of this manuscript is big and vaccination is a part of mitigation measures,

I suggest changing the title of the manuscript to, “Infectious Bronchitis Virus (Gamma-coronavirus) in Poultry Farming: Immune Responses and Mitigation Measures.”

Response: The authors have changed the Title and included your given title as “Infectious Bronchitis Virus (Gammacoronavirus) in Poultry Farming: Vaccination, Immune Response and Measures for Mitigation”.

Also, after the Introduction, I suggest starting with the Immune responses section followed by Potential mitigation approaches in controlling IBV. Vaccine Development (currently section 4.2) needs to be combined with the Vaccination section.

Response: We thank the reviewer for this comment! We also appreciate the given suggestions. All the changes can be found under yellow color changes. We also moved the section “Vaccine Development” under Vaccination as suggested.

Further Comments:

Comments: Line 11 – there is no affiliation #3 in the authors list.-

Response: We are provided all authors affiliation correctly

Citation  - this paragraph refers to another article of this group.  Line 51 – change “SARS-CoV-2 (COVID19)” to “SARS-CoV-2 (etiological agent of COVID19)”.  Line 53 - needs to be “usually infects all kinds of avian species, especially chickens. Lines 55-56 – too many “using” in this sentence. Lines 72-75 – bad English. Line 78 – What is “immune-pathogenesis response” ? Line 84 – words “good feature of IB vaccinations” needs to be changed.  

Response: We are revised the sentences according reviewer comments using yellow color inside manuscript.

Table 1 – needs to be formatted to be on one page. “Delivery system” must be Routes of vaccination. Change “Ovo” to “In ovo”. Viral vectored, recombinant protein (sub-unit) and DNA vaccines must be separated.

Response: We have reformatted the Table 1 according reviewer suggestion

Line 111 – “The difficulties of mild vaccines…” must be “drawbacks” or “disadvantages” . Line 117 – “It has been highly suggested…” may be “strongly suggested”.Line 120 –“ the combination of IBV H52 and H120 vaccine levels” not clear. Line 124-125 – bad English. Line 152-154 – bad sentence. Line 158 – Spell out MDA revise this.  Line 158 -161 – bad sentence. Line 177-178 – bad sentence. Line 185 – “is enclosed in” must be “includes”. Line 191 – “cell-mediated immune responses are protected in chickens” Bad English. Line 206 – “and transaction of adenovirus gene.” What is transaction? (translation).  Line 236 – “in the flock by immunizing monovalent or multivalent in plasmids” Do not understand...  Line 237 – “The finding of results shown…” Bad English... Line 240-244 – Bad sentence. Line 245-247 –Bad sentence. Line 249-252 –Bad sentence. Line 258-259 – Bad sentence. Line 265-267 – Bad sentence.

Response: Adenovirus-based viral vectors most commonly employed used as genetic manipulation means the carrier of transgene DNA insertions. Here, we used transaction is an agreement between adenovirus vector and insertion of IBV S1 gene to express or elicited the transgene-specific T cell and antibody.

We are revised the sentences according reviewer comments using yellow color inside manuscript.

Comments: Figure 2 – I am doubt that production of live vaccine is more time consuming and costly than killed vaccine. What is “interference from normal florea of vaccine (Interference from existing circulatory virueses”? What is “Genetic vaccine”? Is it DNA vaccine, viral vectored vaccine, sub-unit or vaccine generated by reverse genetic technique? Meaning of “limited to protein antigen” is not clear.

Response: Yes, we have mistaken the line in Figure 2 since the author actually make to understand to the cost and time consuming (≥10 years) of IBV live vaccine manufacturer. Thanks for your advice.  Genetic vaccine already divided.

Line 377 – Reference 95 is for human TLRs. What about references for chicken? Line 410 – Mention the main targets for VN antibodies. Line 420 – “days 4-5 after post infection” Line 433-437 – Bad sentence. Figure 4 – Do you have permission to present this figure from reference 27? Please check. Line 516 – “the modern instruments..” ? Line 526 – Bad English. Line 528 – 563 – this section needs to be combined with vaccines. Line 600 – Bad English. Line 610 – Bad English. Line 614 -  “to be stably replicated” ?

Figure 5 – What is the difference between Live vaccine and Live Modified strains?

Response: Our apologies for the errors in the previous Figure 5.  We have modified this Figure (currently Figure 3) and hopefully this will satisfy the reviewer query.  

Reviewer 2 Report

Manuscript "Management of Infectious Bronchitis Virus (Gamma-coronavirus) in Poultry Farming: Vaccination, Immune Response and Measures for Mitigation” (Veterinary Sciences - vetsci-1409419)

Review comments to the authors:

The whole manuscript describes a review about the management of infectious bronchitis virus (IBV). The research question is important. IBV is one of the main pathogens of commercial and backyard chickens, it is disseminated worldwide and vaccination (with an adequate strain) is necessary to control this disease. The message of the whole study is also scientifically interesting, so the manuscript could be published. It is very important to understand whether the introduction of IBV vaccines has the capacity to control viruses circulating in the field.

My main criticism for the whole manuscript is that IBV does not represent a major risk for public health. It is a Gammacoronavirus with many differences from the others genus (Alpha and Beta) that infect humans. Additionally, the current classification of coronaviruses recognizes 39 species in 27 subgenera, five genera and two subfamilies that belong to the family Coronaviridae, suborder Cornidovirineae, order Nidovirales and realm Riboviria.So I think the authors have to delete the word “Biorisk” from the title and to explain better the Coronavirus taxonomy in the Introduction. In addition, it is necessary to remove the topic 4.6: Public health implication. Finally I think it is necessary a topic describing the epidemiology of the disease. It could be the first one after the Introduction, followed by the description of the immune responses and only after the topic about Vaccines.    

There are also some points in the text that can be improved to make the message even clearer:

1)        Please avoid repeating “positive-sense single- stranded RNA virus” and “(+) ssRNA” lines 44-45.

2)        Explain better the Coronavirus taxonomy and the division into the different groups – lines 48-54.

3)        I think the authors should reduce the last paragraph in the Introduction, maintaining only the two last sentences. The remaining of this paragraph should be included in the specific topic number 2.

4)        Please prepare a well-structured Table 1.

5)        May the authors use the images of the Figure 4?

Author Response

Review comments to the authors:

The whole manuscript describes a review about the management of infectious bronchitis virus (IBV). The research question is important. IBV is one of the main pathogens of commercial and backyard chickens, it is disseminated worldwide and vaccination (with an adequate strain) is necessary to control this disease. The message of the whole study is also scientifically interesting, so the manuscript could be published. It is very important to understand whether the introduction of IBV vaccines has the capacity to control viruses circulating in the field.

Response: We thank the Reviewer for the positive comments and appreciate the given suggestions that have been addressed positively.

My main criticism for the whole manuscript is that IBV does not represent a major risk for public health. It is a Gammacoronavirus with many differences from the others genus (Alpha and Beta) that infect humans. Additionally, the current classification of coronaviruses recognizes 39 species in 27 subgenera, five genera and two subfamilies that belong to the family Coronaviridae, suborder Cornidovirineae, order Nidovirales and realm Riboviria.So I think the authors have to delete the word “Biorisk” from the title and to explain better the Coronavirus taxonomy in the Introduction. In addition, it is necessary to remove the topic 4.6: Public health implication. Finally I think it is necessary a topic describing the epidemiology of the disease. It could be the first one after the Introduction, followed by the description of the immune responses and only after the topic about Vaccines.    

Response: We agree with all the suggestions made by the reviewer. Please see the revised manuscript and all the suggested changes can be found under Yellow color.

There are also some points in the text that can be improved to make the message even clearer:

1)    Please avoid repeating “positive-sense single- stranded RNA virus” and “(+) ssRNA” lines 44-45...

Response: Thanks! We have corrected as suggested. 

2)        Explain better the Coronavirus taxonomy and the division into the different groups – lines 48-54.

Response: We have further expanded the taxonomic classification of coronaviruses.

3)        I think the authors should reduce the last paragraph in the Introduction, maintaining only the two last sentences. The remaining of this paragraph should be included in the specific topic number 2.

Response: Thank! To accommodate other reviewer’s suggestion, we have rewritten this section, and deleted some parts that we believe will satisfy this reviewer.  

4)        Please prepare a well-structured Table 1.

Response: We have revised the Table 1 as suggested and included more informations.

5)        May the authors use the images of the Figure 4?

Response: The authors have redrawn the Figure 4. So no copyright issue

Round 2

Reviewer 1 Report

Editing of English language is required. 

Author Response

We have revised the whole manuscript carefully and tried to avoid any grammar or syntax error. In addition, the authors have asked several colleagues who are skilled authors of English language papers to check the English. Now the authors believe that the language is now acceptable for the review process.

Reviewer 2 Report

The authors did a good reviewing job.

Author Response

Again thank you for your realistic comments in our manuscript

This manuscript is a resubmission of an earlier submission. The following is a list of the peer review reports and author responses from that submission.